# An Eye on Kupffer Cells: Development, Phenotype and the Macrophage Niche

**DOI:** 10.3390/ijms23179868

**Published:** 2022-08-30

**Authors:** Andrey Elchaninov, Polina Vishnyakova, Egor Menyailo, Gennady Sukhikh, Timur Fatkhudinov

**Affiliations:** 1Laboratory of Regenerative Medicine, National Medical Research Center for Obstetrics, Gynecology and Perinatology Named after Academician V.I. Kulakov of Ministry of Healthcare of Russian Federation, 117997 Moscow, Russia; 2Histology Department, Pirogov Russian National Research Medical University, 117997 Moscow, Russia; 3Histology Department, Medical Institute, Peoples’ Friendship University of Russia, 117198 Moscow, Russia; 4Laboratory of Growth and Development, Avtsyn Research Institute of Human Morphology of FSBI “Petrovsky National Research Centre of Surgery”, 117418 Moscow, Russia

**Keywords:** macrophages, monocytes, macrophage niche, Kupffer cells, Ito cells, endothelial cells, hepatocytes

## Abstract

Macrophages are key participants in the maintenance of tissue homeostasis under normal and pathological conditions, and implement a rich diversity of functions. The largest population of resident tissue macrophages is found in the liver. Hepatic macrophages, termed Kupffer cells, are involved in the regulation of multiple liver functionalities. Specific differentiation profiles and functional activities of tissue macrophages have been attributed to the shaping role of the so-called tissue niche microenvironments. The fundamental macrophage niche concept was lately shaken by a flood of new data, leading to a revision and substantial update of the concept, which constitutes the main focus of this review. The macrophage community discusses contemporary evidence on the developmental origins of resident macrophages, notably Kupffer cells and the issues of heterogeneity of the hepatic macrophage populations, as well as the roles of proliferation, cell death and migration processes in the maintenance of macrophage populations of the liver. Special consideration is given to interactions of Kupffer cells with other local cell lineages, including Ito cells, sinusoidal endothelium and hepatocytes, which participate in the maintenance of their phenotypical and functional identity.

## 1. Introduction

Macrophages participate in various physiological, immunological and morphogenetic processes including regeneration. Over 95% of all macrophages found in mammalian body are concentrated in the liver, which harbors the most abundant of the resident tissue macrophage populations [1]. Kupffer cells (KCs), the resident liver macrophages, constitute a crucially important component of the mononuclear-monocytic system. KCs have a wide variety of responsibilities at both local and systemic level, notably the barrier function preventing various pathogens and their toxic by-products (e.g., endotoxin, also known as bacterial lipopolysaccharide (LPS)) from entering systemic circulation [2]. KCs constitute estimated 30–35% of the total non-parenchymal liver cell counts [3] and are typically located within the lumina of sinusoidal capillaries adherent to the endothelium, which provides them with immediate access to immunogenic foreign agents that arrive with portal circulation [4]. KCs also participate in protein and lipid metabolism, as well as the clearance of apoptotic cells from circulation [5]. KC dysfunctions have been associated with a number of liver diseases, e.g., viral hepatitis, cholestasis, alcoholic cirrhosis and fibrosis [6]. This review contemplates the role of instructive microenvironment, the so-called tissue niche, in KC functionalities and population dynamics.

## 2. The Origin of Macrophages in Mammalian Ontogeny

It is currently held that mammalian macrophages develop from three sources, corresponding to three generations of hemopoietic stem cells [7,8,9].

The first generation of hemopoietic stem cells emerges in the yolk-sac wall from a subpopulation of mesenchymal cells different from those giving rise to endothelium of the primary capillaries [8,9]. This first generation is thought to give rise to highly specialized macrophage lineages of the central nervous system (CNS)—the microglia [10,11]. Microglia is considered unique among macrophage lineages in not having the monocyte stage: microglial precursors are forwarded to CNS very early in embryogenesis [10,11].

The second generation of hemopoietic cells, known as erythro-myeloid progenitor cells, is derived from the hemogenic endothelium of the yolk-sac capillaries. Although macrophages differentiating from these precursors have molecular phenotypic signatures very similar to those of the first generation macrophages, the differentiation already proceeds through the monocyte stage [8,9,10,11]. This generation of erythro-myeloid cells gives rise to most resident macrophage populations, including KCs.

The third generation of hemopoietic progenitor cells develops from the hemogenic entothelium of the aorto-gonado-mesonephral zone to further colonize the liver, the bone marrow and other embryonic organs except CNS [10,11].

Thus, the resident macrophage populations of organs are initially formed by descendants of hemopoietic cells of the second and third generations. In most organs, however, the fraction of macrophages descending from erythro-myeloid progenitor cells of the yolk sac eventually declines, whereas the fraction of third generation hemopoietic cell-derived macrophages prevails [7,8,9]. The exceptions include CNS (untresspassable to macrophages apart from the first generation-derived microglia), the liver and the epidermis (normally harboring only second generation-derived macrophages—Kupffer cells and Langerhans cells, respectively [8,9].

During postnatal period, many resident macrophage populations tend to be replaced with newly arriving, bone marrow-derived monocytes. The extent of such replacement depends on the particular organ and pathophysiological circumstances (inflammation status, regeneration, etc.) and is subject to further specification. Liu et al. (2019) [12] identify *Ms4a3* gene expressed by common granulocyte-monocyte progenitors in the bone marrow and use it as a marker to trace the migration paths for granulocytes and monocytes. The data obtained with this new marker confirm the earlier findings that at many locations the resident macrophage populations exist independently of bone marrow hematopoiesis. A perfect example of such a location is the liver, where the percentage of ‘truly resident’ macrophages is extraordinary high—estimated at 92%. According to the Ms4a3 gene expression dynamics, the rates and extent of the resident macrophage replacement with monocytic macrophages depends on the type (model) of inflammatory reaction and is likely to be organ specific [12].

It should be noted that macrophage ontogeny has been predominantly studied in murine models. However, the routes of hemopoiesis are thought to be highly conserved and universal among mammals [13,14,15]. Consistently with this assumption, new data on macrophage ontogeny in humans indicates its generic similarity to rodent models, despite the species-specific differentiation scopes and topographical ranges of certain progenitors [13,16].

## 3. Monocyte as Archetypal Stage in Macrophage Development

Monocytes are white blood cells emanating from the red bone marrow as precursors for macrophages and dendritic cells in peripheral tissues. Monocytes belong to the mononuclear phagocytic system and their circulating pools are constantly replenished. Blood monocytes descend from the common myeloid progenitor (CMP) cells located in the red bone marrow [17]. CMP has the potential to differentiate into megakaryocyte and erythrocyte progenitor or granulocyte-and-macrophage progenitor (GMP) [18]. The origin of one of the next monocytes’ ancestors, the monocyte/DC progenitor (MDP), is a matter of debate [17]. On the one hand, MDP is considered the result of GMP lineage commitment [19], and on the other hand, these cells were also found among CMP compartment [20]. Moreover, Yanez et al. showed that both Ly6Chi and Ly6Clow monocytes could appear simultaneously from two routes of monocyte differentiation: from GMP and the MDP population [20]. MDP population, in turn, gives rise to common DC precursors (CDP) and unipotent common monocyte progenitors (cMoP). cMoP has high proliferative capacity and is characterized by the expression of common monocytes’ markers: Ly6C in mice and CD14 in humans [21,22]. Before becoming mature monocytes, cMoP undergoes a transitional premonocyte (TpMo) stage whose phenotype and transcriptional signature was well-described by Chong et al. [21]. The TpMo precursor is characterized by a high expression of CXCR4 and CD31, which during the course of monocyte maturation decreases, whereas CCR2, CX3CR1 and CD11b become upregulated. Following the egress from the red bone marrow, a monocyte circulates for several days and dies by apoptosis unless infiltrated into a tissue.

Monocytes are heterogeneous cells, and with the development of multicolor cytometry, the range of classifications of their subpopulation composition is expanding. The established classification of monocyte subtypes divides them into a classical population (Ly6Chi in mice; CD14++CD16− in humans) and a non-classical population (Ly6Clo in mice; CD14+CD16++ in humans) [17]. An intermediate subpopulation is uniquely identified in humans (CD14++CD16+) and in mice according to a number of authors [23,24,25]. At the moment, it is believed that the population of non-classical monocytes descends directly from the classical ones [25,26,27]. Experiments with BrdU labelling revealed that Ly6Chi monocytes constitute obligatory steady-state precursors of blood-resident Ly6Clo cells. In addition, convincing evidence on Ly6Chi conversion into Ly6Clo monocytes based on data of chromatin analysis was shown in the work of Mildner and colleagues [27]. The potential of classical monocytes to give rise to intermediate and nonclassical monocytes was also shown for human cells [28]. It has been shown that BrdU-labeled Ly6Chi monocytes circulate in the bloodstream for 5 days and then begin to differentiate into Ly6Clo monocytes [27]. Obviously, a part of classical monocytes will be infiltrated into the tissue in response to signals, since monocyte subpopulations are characterized not only by phenotypic but also by functional heterogeneity. Classical monocytes are thought to be inflammatory cells with high tissue infiltrating capacity. Upon detection of an attractive signal from the tissue, classical monocytes migrate to the source and differentiate into macrophages, which are often functionally distinct from resident macrophages of embryonic origin. The phenomenon in which monocytes lose their monocytic phenotype after tissue infiltration in the intestinal context is called a “monocyte waterfall” [29,30]. However, this is not a general rule, since monocytes are able to maintain their monocytic character even after infiltration into the tissue, which was shown in a remarkable work by Jakubzick and colleagues [31]. It is believed that the molecules that attract monocytes to infiltrate tissue are mostly chemokines (e.g., CCL2/MCP-1, CCL7/MCP-3, CX3CL1) [32,33,34,35,36].

In contrast, nonclassical monocytes seem to be patrolling cells that remain in the circulation and patrol the vessels to scavenge them and maintain the integrity of the endothelium [25,37]. It has now been established that nonclassical monocytes are the first to migrate into tissues in response to inflammatory signals and are the source of the synthesis of the first portions of TNFa and IL1 necessary to trigger the inflammatory cascade and the migration of classical monocytes and other leukocytes, but do not differentiate into macrophages [38].

## 4. Macrophage Populations of the Liver

An accumulating body of evidence reveals extraordinary heterogeneity of liver macrophages, explained by multiple intersecting diversities (diverse sources of origin, diverse hepatic functionalities and diverse pathophysiological circumstances) [39].

According to the current state of knowledge, a normal liver harbors at least three populations of cells constituted by monocytic-macrophage lineages: (1) the dominating KCs; (2) cells with intermediate phenotypes between monocytes and macrophages; and (3) non-KC macrophages, including liver capsule macrophages, peritoneal macrophages and biliary tree-associated macrophages [40] resembling the Gpnmb+Spp1+ lipid-associated macrophages, LAMs [41].

In terms of embryonic origin, liver macrophage populations almost entirely descend from the erythro-myeloid progenitors of the yolk-sac wall [9,11]. These progenitors migrate through the left vitelline vein and umbilical vein to the embryonic liver, colonize it and embark on differentiation into KCs till later stages of fetal development and after birth [42]. Macrophages derived from blood monocytes constitute a minor fraction within the liver, commonly estimated within 10% of total liver macrophages and probably accounting for up to 30% in certain models [7,43,44].

Considering the diversity of embryonic sources and to avoid confusion, many authors stick to the following nomenclature of liver macrophages. The term ‘Kupffer cells’ is assigned exclusively to macrophages descending from the erythro-myeloid sources of the yolk-sac wall [45,46]. However, other authors use this term less scrupulously to refer to all macrophages found within the liver independently of their origin. Authors of current review favor the first option, as it really allows avoiding confusion while acknowledging the diversity of embryonic sources contributing to the hepatic macrophage populations.

Estimates of the proportion of bone marrow-derived (monocytic) macrophages in the liver vary greatly due to the lack of unified set of markers [47,48]. In our opinion, the most accurate estimates are obtained with Ly6C and CX3CR1 [7,8,9,44,49].

Most studies of macrophage ontogeny have been performed on laboratory mice. Murine macrophages of bone marrow origin, in postnatal development, express Ly6C protein on their surface, whereas KCs either lack this marker or express it at low levels [44,50]. In rats, Ly6C protein has not been identified and CX3CR1 is used as a protein marker of bone marrow-derived (monocytic) macrophages instead of Ly6C [44]. As estimated with these markers, about 5% of total liver macrophages originate from the bone marrow [7,8,9,44,49].

These estimates are considered with morphological data on the dimensional and topographical diversity of liver macrophages. According to these observations, liver macrophages fall into two morphological subtypes: ‘large’, associated with sinusoidal capillaries, and ‘small’, located in the vicinity of central veins and portal tracts. Both subtypes express CD68, but only ‘large’ macrophages express high levels of CD163 [51,52]. Furthermore, ‘small’ macrophages constitute about 8% of the total liver macrophage counts, which matches the estimates of monocytic macrophage content obtained using Ly6C marker [7,44]. Hence, the populations of ‘small’ liver macrophages and liver cells of bone marrow origin may prove to be the same.

Although many studies demonstrate high expression levels of mannose receptor protein CD206 by resident liver macrophages, dedicated analysis shows that the degree of CD206 positivity varies. Accordingly, KCs can be subdivided into two subsets: a predominant CD206loESAM- (KC1) and a CD206hiESAM+ minority (KC2) [53]. The KC2 cells have been shown to express genes that regulate fatty acid metabolism under normal and pathological conditions; moreover, KC2, which express high levels of CD36, have been shown to participate in regulation of the obesity-related oxidative stress in the liver. In addition, KC2 cells under the action of IL-2 promote CD8+ T cell activation thereby supporting the antiviral immunity [54].

The most controversial findings so far have been obtained with CD11b and CD68 markers expressed by a wide variety of cell types including all leukocytes; in addition, CD68 is expressed by endothelial cells and fibroblasts [55]. CD11b and CD68 participate in cell adhesion, migration and phagocytosis; accordingly their expression can undergo rapid changes [55,56,57]. The use of CD11b and CD68 complemented with F4/80 helps identify at least three F4/80+ liver macrophage subpopulations with distinct functionalities: cytokine-producing F4/80+CD11b+, highly phagocytic F4/80+CD68+ and F4/80+CD11b-CD68- with yet unknown function. CD11b+ liver macrophages presumably contribute to anti-tumor immunity [58] (Table 1).

Some studies use CD11b as a marker of bone marrow-derived (monocytic) macrophages [43,65]. Importantly, although CD11b is expressed by both KCs and monocytic macrophages of the liver, monocytic macrophages express it at higher levels than KCs [59,60].

A remarkable series of recent studies introduce a special macrophage population inhabiting the connective tissue capsule of the liver [62,63]. It is reasonable to assume that KCs associated with sinusoidal capillaries eliminate pathogens that have already entered the blood, but obviously cannot interfere with the spread of pathogens in the abdominal cavity. The authors describe a population of macrophages within the liver capsule that differ from KCs both phenotypically and by origin, which probably participate in the intra-abdominal clearance, showing that up to 3 × 10^5^ macrophages inhabit a single mouse liver capsule [62,63]. The liver capsule macrophages (LCMs) develop from bone marrow precursors and have very long processes (a morphological distinction). The data indicate that LCMs express F4/80 as well as other macrophage markers such as CD64, CSF-1R, CX3CR1 and CD14. At the same time, they express CD11c at low levels and express neither CD103 nor Tim4. The main function of LCMs is to defend the liver from microbial pathogens [63], partly by stimulating the neutrophil infiltration of the liver. Although LCMs express CD207, similar CD207+ macrophages can be observed in the vicinity of central veins in liver lobules [40,63].

Another recently described liver macrophage population is associated with bile ducts and is positive for the Gpnmb marker, which identifies them as lipid-associated macrophages (LAMs) apparently derived from recruited monocytes [41]. LAMs are less sensitive to endotoxin/LPS than KCs, which may be due to their affiliation with the branching portal vein delivering LPS and other pathogen-related products to the liver, and so LAMs may develop tolerance to these agents [41]; there is no decisive evidence on this subject as yet.

The enhanced heterogeneity of liver macrophages under pathological conditions reflects both the immigration of monocytic macrophages and the phenotypic alterations in KCs.

Murine model of steatohepatitis harbors two major populations of liver macrophages: KCs expressing Clec4f and monocytic macrophages expressing Lyz2 [66]; at that, the population of monocytic macrophages is heterogeneous in itself. The authors identify at least three cell subtypes within monocytic macrophage population: (I) expressing high levels of Fn1, Mgst1 and Msrb1; (II) expressing high levels of Chil1; and (III) expressing Il1b [66].

## 5. Components of the Liver Macrophage Population Dynamics: Cell Migration, Cell Proliferation and Cell Death

The foundations of contemporary views of mononuclear-phagocytic system were laid in the 1970s by the works of Ralph van Furth. He suggested a concept of continuous replacement of tissue macrophages (with a naturally limited life span) by immigrating monocytes originating in the red bone marrow and transported by circulation. As the highly differentiated resident macrophages were assumed to lack the mitotic capacity, this view was fully justified and consistent with certain experimental findings [67]. Nevertheless, it eventually became evident that a majority of resident macrophage populations, including KCs, are self-perpetuating rather than dependent on the bone marrow hemopoiesis [68]. At the same time, the canonical macrophage differentiation scheme involving blood monocytes is possible as well and actually dominates in many postnatal tissues (Figure 1). For the liver, the canonical scheme is used as an emergency fallback, i.e., the infiltration of the liver with monocytes can be observed under pathological conditions only [69], whereas physiological contribution of monocyte immigration to the liver macrophage counts is negligible.

The remarkable proliferative capacity of liver macrophages, most unexpected by the researchers, turned out to be virtually the only route of the liver macrophage population maintenance under physiological conditions and even in certain pathologies [68].

It should be noted that in some studies, even after non-genotoxic depletion of local macrophages in lungs and red bone marrow, their populations are successfully replenished by means of resident cell proliferation with minimal participation of bone marrow-derived monocytic precursors. However, with the temporary block of the local macrophage proliferation (e.g., by applying a lethal dose of ionizing radiation), the population is restored by recruitment of circulating monocytes [70,71]. Macrophage proliferation is presumably stimulated by IL-4 acting independently of the cell origin (monocytic or resident) [68,72].

The proliferative capacity specifically of KCs has been demonstrated experimentally in various models, including resections of different volume and acute liver injury [44,73]. For instance, after 70% liver mass resection in mice, up to 50% of macrophages within the liver remnant enter proliferation [74]. Importantly, for Kupffer cells, the main proliferation-driving cytokine is not IL-4, but IL-6 [75], which indicates some organ-specificity of mitotic cycle regulation mechanisms in macrophages. For lung macrophages and some other macrophage populations, proliferation has been shown to depend on M-CSF and GM-CSF as well [70].

The proliferative capacity of KCs has clinical significance. For instance, in non-alcoholic steatohepatitis, the mitotic capacity of KCs is blocked, and their population is replaced with monocytic macrophages that gradually acquire KC-like phenotypes. However, these ‘new’ macrophages have pro-inflammatory phenotypes and metabolize triglycerides less efficiently, which eventually leads to aggravation of the condition [76].

Another component process affecting macrophage numbers in the liver is migration of blood monocytes to the liver and their subsequent differentiation into macrophages. This component is most pronounced under conditions of toxic liver injury modeled in laboratory animals with the use of carbon tetrachloride or acetaminophen (paracetamol). The numbers of immigrating monocytes in the liver peaked 24 h after acetaminophen-induced damage; the cells differentiated into macrophages and were subsequently eliminated from the liver [44].

The immigration of monocytes/macrophages, albeit at a much lower scale, can be observed during regeneration of the liver after 70% resection in mice. An increase in Ly6C+ and CD11b+ cell numbers within the liver as early as 24 h after resection was demonstrated. Notably, the counts of Ly6C+ cells continue to increase until day 7 post-resection, amid a decline in CD11b+ cell counts [74]. The extent of monocyte/macrophage migration to the remnant liver depends on both the resection volume and model animal species. For example, resection of more than 80% liver volume in rats promotes negligible migration of monocytic macrophages to the remnant liver [73].

It has long been held that macrophages immigrating to the liver are totally derived from blood monocytes. This long-standing opinion was challenged by recent findings obtained in various in vivo settings, including hepatotoxicity models and 70% liver resections in mice.

Apart from the blood monocytes, an alternative source of macrophages immigrating to the liver can be provided by peritoneal macrophages [64]. This phenomenon was discovered in a model of localized sterile heat injury of the liver: macrophages migrating to the area of damage expressed classical markers but also CD102 and GATA6 indicating their belonging to the peritoneal macrophage population. Experimental findings by another group of authors indicate that this population does not penetrate deep into the liver and is not involved in the inflammatory and repair processes [77]. It should be noted that macrophages with F4/80+Ly6C+CD11b+ phenotype, corresponding to peritoneal macrophages, also appear in the liver after resections, but their origin remains unexplored [64,74,78].

One of the main uncertainties concerning liver macrophage populations under pathological conditions is their further destiny. Several studies show that after the completion of repair processes in the liver monocytic macrophages become eliminated, whereas the resident macrophage numbers are restored by means of resident macrophage proliferation [44,50], and such scenario has been demonstrated for inflammatory damage in other organs as well [79]. However, with depletion of KCs from the liver, the niche is filled by immigrating monocytic macrophages which successfully engage in the long-term maintenance by proliferation [61,80], although in other organs recolonization proceeds differently for reasons as yet unexplained [70].

Moreover, the degree of identity of the colonizing monocytic derivatives to KC is rather controversial. Some studies demonstrate full correspondence between the two lineages, both molecular (gene expression profiles) and functional (proliferation and phagocytosis capacities) [61]. Other studies show that the differences persist: for instance, KC express Tim4 and Marco at higher levels and engulf acetylated low density lipoprotein with higher intensity, while showing lower rates of phagocytosis towards bacterial pathogens, as compared to the substitute bone marrow-derived macrophages [81]. The controversy possibly results from different observation lengths used in the studies and can be interpreted as follows: the longer bone marrow-derived macrophages stay in the liver (depleted of KC) the closer they come to resemble KC phenotypically and functionally [39]. This issue will be given additional consideration in an upcoming section on the macrophage niche of the liver.

Cell death is yet another component process affecting liver macrophage ‘demography’ and the mechanisms of its regulation are poorly understood. Relatively few studies have been focused on this issue, although many studies feature the massive death of macrophages within the framework of primary tissue response to damage, and the liver is no exception. The death of resident macrophages qualified as necroptosis or necrosis is typical for bacterial and viral infections, as well as malaria [82,83,84]. This phenomenon, described for the first time in alveolar macrophages and termed ‘defensive suicide’, essentially triggers the defensive inflammatory reaction and is by no means a passive casualty of the invading microbial pathogen [83,85]. Increased rates of cell death in liver macrophages have been described in various murine models (hepatotoxic injury, 70% liver resection), with the scale of cell death among F4/80+ liver macrophages reaching 16% [74]. The scale, functional significance and molecular mechanisms of macrophage cell death during tissue repair, particularly in the liver, remain understudied. Clearly, the death of resident macrophages may represent a part of tissue response, acting as a trigger for repair processes.

## 6. KC-Specific Phenotypes

Gene and protein expression profiles identified with KCs are fairly flexible and depend on both the paracrine landscape and the long-range signals arriving from other organs [86]. Despite their pronounced molecular plasticity, KCs have been attributed with a particular immunophenotype [81,87]. Its hallmark proteins MARCO and CD163 are responsible for the recognition of bacterial pathogens and triggering of local immunity reactions; another KC marker, CD206, participates in antigen presentation, phagocytosis, cytokine production and pro-inflammatory mediator clearance [88]. In addition, resting KCs express the so-called tolerogenic phenotype variation with characteristic transcriptomic signature [60] including elevated expression of vascular permeability factors, ion channels, hemoglobin metabolism and complement system genes [81]. Such tolerogenic phenotypes are necessary for the active suppression of immune responses to the continuous influx of immunogenic agents accompanying the absorbed semi-metabolized nutrients and tissue debris [4,89].

Accordingly, KCs utilize enormous quantities of endotoxin/LPS without promoting inflammatory reactions. Moreover, LPS seems to trigger anti-inflammatory activation of KCs. Under these conditions, KCs suppress activation and proliferation of helper T cells while attracting regulatory T cells [90,91]. In addition, LPS may facilitate Fas-L expression by KCs, which triggers apoptosis in T cells [92]. At the same time, depletion of KCs causes fatal outcomes in bacterial infections with *Listeria monocytogenes*, *Brucella burgdorferi* or *Staphylococcus aureus* [93,94,95]. 

KCs also express the immunomodulatory PD-L1 and PD-1 shown, for example, to suppress the activity and proliferation of killer T cells in hepatocellular carcinoma or chronic hepatitis B [96]. The inhibition of PD-L1 expression by KCs in CMV infections stimulates the antiviral immunity [97]. KCs have been also shown to produce prostaglandins PGE2 and 15d-PGJ2 that interfere with the antigen-specific activation of T cells [98]. The loss of tolerogenic phenotype by Kupffer cells facilitates the development of inflammatory processes not only in the liver, but outside it as well [90].

Clec4F and Tim4 proteins and their corresponding transcripts are hallmarks for KC-specific molecular signatures [61]. Clec4F is a C-type lectin participating in antigen presentation of glycolipid antigens, as well as in the recognition and scavenging of desialylated platelets [99,100]. Tim4 is a phosphatidylserine-specific receptor allowing KCs to scavenge dying cells, also involved in triggering Th2 cell differentiation. Experimental inactivation of Tim4 improves the engraftment of the liver in allogeneic transplantations [101,102].

Overall, the diversity of KC phenotypes and gene expression signatures reflect the functional diversity characteristic of these liver-specific resident macrophages: KCs effectively combine the conventional macrophage functionalities, such as antigen presentation and phagocytosis, with organ-specific chores that involve elimination and scavenging of senescent formed elements, as well as tolerogenic influences.

## 7. The Concept of Macrophage Niche and Its Application to KCs

### 7.1. The Concept of Macrophage Niche

The history of studies on the macrophage system in mammals is fairly long. The accumulating evidence has increasingly suggested that macrophages are extremely heterogeneous both phenotypically and functionally and thus must not be regarded as a single, uniform population [103]. Though specific prerequisites for such heterogeneity are uncertain, it apparently reflects the diversity of tissue microenvironments where macrophages differentiate and which are thought to largely define their phenotypes and functionalities [86]. 

These considerations eventually crystallized into the macrophage niche concept putting an emphasis on the unity of cellular components, extracellular matrix (ECM) and biologically active signaling molecules in providing the immediate environment for the tissue macrophage maturation [104]. The macrophage niche components ensure spatial compartmentalization/scaffolding and implement the trophic function; most notably in the context of this review, they provide macrophages with tissue-specific cellular identity by inducing particular sets of key transcription factor genes [80,105]. 

The macrophage niche concept is intended not just to explain the diversity of macrophage phenotypes and functionalities but also, very importantly, to account for the full-scale preservation of ‘relict’ macrophages of early hemopoietic origin in many organs and their non-replacement (resistance to replacement, as shown by experimental research) with descendants of fresher hemopoietic lineages [86,104]. The primary mechanistic explanation for this state of events in terms of macrophage niche involved three parameters: accessibility, vacancy and competition for the niche [86,104]. Accordingly, determination of macrophage composition at a particular location in the body could be reduced to interplay of these parameters. For instance, the persistence of microglia as the unique macrophage population in CNS could be explained by the non-accessibility of this location to monocytic precursors, due to the presence of the blood-brain barrier [86,104]. In contrast with the brain, the liver is accessible throughout postnatal development; however, by the time of full-fledged hemopoiesis in the bone marrow, all hepatic macrophage niches turn out to be occupied; upon injury, some of these niches go vacant which enables the immigration of monocytes from the blood and their subsequent differentiation into macrophages. In the lungs, the macrophage niches are constantly accessible and some of them vacant (free niches constantly appear physiologically), albeit on competitive terms, which explains the prolonged coexistence of macrophages from different sources in the lungs, with a gradual increase in the share of bone marrow-derived lung macrophages during postnatal life [86]. 

At the same time, some experimentally observed features of local macrophage populations in mammalian organs are only partially consistent with the fundamental macrophage niche concept. For example, monocytes arriving in the lungs are capable of differentiation into macrophages indistinguishable from the resident [106]. In a similar study, the liver depleted of resident macrophages was colonized by monocytes of bone marrow origin arriving from the blood. The macrophages differentiating from these monocytes were functionally similar to KC, but expressed a different profile of transcription factors [61,81].

Furthermore, a straightforward implementation of a lung-like scheme of macrophage colonization in the liver would imply a gradual increase in the content of bone marrow-derived macrophages as the liver grows. In laboratory rodents, the liver grows continuously throughout life, but the proportion of bone marrow-derived liver macrophages stays low—reaching 2–5% soon after birth and remaining at this level later on [8,9].

Complete ousting of the newly arrived bone marrow-derived (monocytic) macrophages by proliferating resident macrophages of the liver was demonstrated in a murine model of toxic liver injury [50]. A similar wave of monocyte immigration to the liver is observed after 70% liver resection in mice (although the fate of these cells has not been studied) but intriguingly not after subtotal resection (over 80% of the organ volume). According to the basic concept, the appearance of new macrophage niches during organ growth should promote monocyte/macrophage immigration as the cheapest route of replenishment [86]. The negligible rates of monocyte immigration in murine subtotal liver resection model may be related to low levels of MCP-1 production in the remnant [107].

The accumulation of new findings has fostered an attempt to modify the macrophage niche concept (Figure 2). The authors dismiss the assumption that each organ comprises a single macrophage niche as inaccurate, and such opinion is consistent with representation of multiple macrophage lineages, including KCs, liver capsule macrophages and probably also peritoneal macrophages, in the liver [64]. A similar situation has been described for other organs, e.g., lungs, and even CNS microglia shows region-dependent heterogeneity [108,109]. Special types of macrophage populations have been identified in the so-called border zones, for example, the already mentioned macrophage population of the liver capsule, biliary tree-associated macrophages or macrophages found in the vicinity of mammary ducts [40,110].

Another consideration to be added to the fundamental macrophage niche concept is a new parameter denoted as “time of residence” within the organ. Introduction of this parameter was necessitated by apparent controversy of experimental findings—success or failure to distinguish between true resident macrophages and those differentiated from the arriving blood monocytes depending on particular experimental setting [61,81]. Given that in many organs resident macrophage populations are gradually replaced by bone marrow-derived cells, such mixed populations are significantly heterogeneous in terms of time since colonization, which can be also defined as the niche occupation length. Expression levels of TIMD4 (TIM4) protein have been shown to correlate with the length of macrophage lineage affiliation with its current place of residence (niche). For instance, Scott et al. (2016) observed rapid (in the course of several days) acquisition of KC-like phenotypes by monocytes arriving in the liver; at that, *Timd4* expression in these new KC-like monocytic macrophages stayed negligible and its induction was delayed for over a month [61]. These data are consistent with the results of studies on liver macrophage dynamics in non-alcoholic steatohepatitis murine model [41].

To take stock, the process of full differentiation of the newly arrived monocytes into resident liver macrophages consists of two stages. At the first of them, a vacant macrophage niche generates ‘stay here’ signals to facilitate quick adaptation; the second stage, assimilation, is lengthier: the macrophage becomes fully integrated in the niche, receiving support in the form of ‘learn this’ signals [80,111].

Despite the opinion that monocytes can be differentiated in any type of tissue macrophages and exactly mimic any resident macrophage lineage except CNS microglia [105], monocyte-derived mouse liver macrophages still differed by their expression profiles from KCs as late as 6 weeks post-colonization [81], and a similar delay in leukocyte differentiation apparently occurs in the human liver [112].

To what extent does TIM4 expression reflect the length of stay and the depth of monocytic macrophage assimilation in the macrophage niche of KCs? In our opinion, this is a complex issue. Our own data obtained in a model of 70% liver resection in mice indicate that *Tim4* expression in hepatic tissues can rapidly increase in the remnant liver tissue. This increase can be attributed to the influence of boosted endotoxin/LPS blood levels, and this suggestion has been supported by in vitro experiments [113]. Notably, KCs isolated from the intact liver by magnetic sorting for F4/80 had *Tim4* expression levels similar to macrophages differentiated from peripheral blood monocytes isolated by magnetic sorting for CD115 and cultured with M-CSF. Under LPS exposure, *Tim4* expression was significantly upregulated in both types of cultures. As it was also demonstrated, after liver resection in mice Ly6C+ monocytes migrate to it in high numbers. Considering the reduced *Tim4* expression in the ‘newcomers’ [61], it would be reasonable to expect decreased expression of TIM4 marker in total liver macrophages after resection. However, this proved not to be the case, apparently due to the stimulating effect of LPS on the new, differentiating monocytic macrophages [113]. It can be concluded that, in this setting, *Tim4* expression levels depend more on LPS exposure than on the length of residence in the macrophage niche of KCs [113].

Similar considerations and findings are applicable to another marker of resident liver macrophages, MARCO [81]. The use of this marker for distinguishing between macrophages arising from different sources has been based on the assumption that its expression is relatively constant and does not respond to endotoxin/LPS [81]. However, the latter point is questionable, considering the long-known role of this receptor in antimicrobial immunity [114,115]. True enough, KCs and bone marrow-derived macrophages express *Marco* at different levels; however, a sharp increase in *Marco* expression at both mRNA and protein levels in the liver remnant was observed after 70% hepatectomy, and this effect was reproduced by in vitro exposure of liver macrophage cultures to LPS [113]. Such dynamics of MARCO expression are consistent with its being a marker of pro-inflammatory state in murine macrophages [103].

The endotoxin/LPS sensitivity of the candidate macrophage markers is essential, as the vast portion of LPS in mammalian body is metabolized by the liver, not to mention the role of any tissue macrophages in antibacterial defense [116]. These considerations further implicate the tissue inflammatory status as a decisive factor which determines the phenotypical and functional properties of macrophages. The impact of inflammatory status on macrophage functionalities is especially prominent in the liver, given the barrier function of the organ. It has been already mentioned that various infections can promote a wave of cell death in resident macrophages, considered a ‘defensive suicide’ [85]. It is important to note that infectious lesions, which become sites of massive death of resident macrophages, are rapidly colonized by macrophages differentiating from the migratory blood monocytes. Notably, during influenza A infections, the resident alveolar macrophages become replaced by monocyte-derived macrophages, which appear to be more efficient in fighting *Streptococcus pneumoniae* infections due to higher production levels of IL6, CCL3, CCL4 and G-CSF [117,118]. Similarly, after a herpesvirus infection, alveolar macrophages effectively prevent the development of asthma by virtue of replacement of the ‘old’ resident alveolar macrophages by ‘new’ regulatory monocyte-derived macrophages that block the ability of dendritic cells to trigger Th2 responses [119].

A team of authors observing an extraordinary robust immune response to *S. pneumoniae* following adenoviral infection attributed it to a previously unidentified ‘macrophage memory’ phenomenon, thereby suggesting the existence of memory cells in macrophage lineages [117,118]. In essence, the adenoviral infection causes activation of the resident alveolar macrophages, thus stimulating formation of a special self-perpetuating alveolar macrophage population. Upon activation, the naïve alveolar memory macrophages memorize the microenvironmental cues under the influence of Th cells and IFN-γ released by them [118]. The identified memory macrophage population is considered self-perpetuating and independent of blood monocytes. These cells are believed to retrieve the memorized information on previous inflammatory reactions to microbial pathogens in case of new infections; their action involves recruitment of neutrophils to the inflammatory foci [118].

The monocyte-derived macrophages which colonize the liver in the aftermath of KC depletion provide more efficient clearance of *Neisseria meningitidis* or *Listeria monocytogenes* by phagocytosis compared with KCs [81]; they also exert a more pronounced pro-inflammatory effect [76]. Our own in vitro experiments have demonstrated that, at early time points of stimulation, monocytic macrophages engulf latex particles at higher rates compared with KCs [87,120]. Such data can be interpreted in terms of hepatic tissue macrophage niche and its special features.

Over the entire history of research on the mononuclear phagocyte system in mammals macrophages were considered as cells with pronounced phenotypical plasticity confirmed in numerous studies [103]. Comparative evaluation of resident macrophages vs blood monocyte-derived macrophages shows higher sensitivity of the latter to activating factors and their higher phenotypic plasticity. Presumably, the prolonged exposure of resident macrophages to the conditions of organ-specific tissue niche leads to a reduction in plasticity through epigenetic block of inflammation-related genes. Such suppression is beneficial, as it suits the needs of the organ homeostasis. This view is based on research involving alveolar macrophages [79] and its validity for other resident macrophage populations, including those of the liver, has not been verified so far.

The liver has long been considered an immunotolerogenic organ [4]. Maintenance of this capacity is largely a responsibility and merit of KCs. Under normal conditions, KCs produce PD-L1, which participates in suppression of cellular immunity reactions, and small amounts of TNFa and IL-12. Under stimulation, KCs produce both pro-and anti-inflammatory cytokines [60,121], but their responses to many pathogen-associated molecular patterns (PAMPs), notably those of endotoxin/LPS are remarkably low. One of the probable reasons is the continuous exposure of KCs to LPS, the concentration of which in the portal blood flow varies within 0.1–1 ng/mL [116]. Such exposure presumably endows KCs with LPS tolerance, or at least reduced LPS sensitivity, compared with blood monocytes and monocyte-derived macrophages. Our own data agree with this assumption. For example, *Tlr4* expression in peripheral blood monocytes is significantly higher compared with KCs [122,123], which is consistent with the evidence on more facile and LPS-sensitive induction of synthesis of certain interleukins in monocytic macrophages [87]. At the same time, studying the expression of LPS tolerance-associated genes in KCs and monocytic macrophages, no classical signatures of LPS tolerance were found. Still, KCs revealed lower expression of MAPK signaling-related genes Erk2 and p38 [124], known to participate in pro-inflammatory cytokine synthesis and release by macrophages. Reduced production of cytokines by macrophages has been associated with tolerance [125,126].

It can be assumed that similar mechanisms can reduce KC tolerance to PAMPs other than endotoxin/LPS: for instance, higher expression of Tlr2, Tlr7 and Tlr8 genes in monocytes compared with KCs was observed [122]. It should be also noted that the benefit of re-colonization with monocytic macrophages with regard to prevention of asthma and resistance to bacterial pathogens observed in the lungs cannot be straightforwardly extrapolated to the liver. A similar replacement scenario applied to KCs in the liver may have a deeply damaging effect and result in a chronic inflammatory process in hepatic tissues and at systemic level [76,90].

The introduction of the tissue inflammatory status as one of the parameters ‘in charge’ of the counts and properties of macrophages in particular organ may also ease the apparent controversy concerning the cell fate of monocytes colonizing the liver after toxic injury. As has been mentioned, these monocytes colonize the injured liver to become totally eliminated later on, despite a sharp decline of the resident macrophage populations and the overwhelming abundance of vacant niches. Presumably, the inflammatory status, and notably the time window of elevated TNFa and IL-1 levels determine the permission for monocytes to occupy the vacant niches previously occupied by KCs [80,82]. Still, even under this assumption, it is difficult to explain subsequent disappearance of these new monocytic lineages after resolution of the inflammatory process, especially given that at early stages of recovery the immigrating monocytes outweigh the preserved KCs numerically. For the reasons as yet unknown, in the toxically injured liver, surviving KCs clearly outcompete the immigrating monocytes/macrophages in settling the vacant niches. Incidentally, despite the just-experienced toxic shock, resident macrophages enter proliferation much earlier. The comparative dynamics imply that to make the monocytic impact visible, 80% of the resident liver macrophages should be depleted, which is hardly possible to achieve with available experimental techniques [80,105].

Notably, in the rat model of subtotal live resection, expression of *Tnfa* and *Il1* increased significantly by the end of regeneration only, whereas the content of TNFa protein in the remnant was reduced from the beginning and stayed low since. This observation can be related to the lack of immigration of CX3CR1+ macrophages to the remnant liver in this model [73,127].

With these important amendments to the fundamental macrophage niche concept, the macrophage populations of individual organs are perceived as complex systems engaged in specific functionalities depending on their localization and ensuring the communication among different compartments inside each organ [105]. Moreover, macrophages have been implicated in the inter-organ crosstalk as well [39]. For example, myocardial overload leads to activation of the sympathetic innervation of the kidneys, resulting in enhanced secretion of S100A8/A9 peptides by the collecting duct epithelium. These peptides stimulate kidney macrophages to release TNFa, which promotes secretion of GM-CSF by endothelial cells in the interstitium. In return, the increased blood levels of GM-CSF promote accumulation of Ly6Clo macrophages in the myocardium; these macrophages produce amphiregulin, which causes hypertrophy of cardiomyocytes [128]. Similar data on the relationship between macrophages of the heart, lungs and kidneys were obtained in a model of myocardial infarction [129].

This new paradigm provides an unexpected explanation to the experimentally observed increased expression of certain interleukins and growth factors in lungs and kidneys after subtotal hepatectomy in rats [127,130]. The effect is accompanied by an increase in CD68+ macrophage counts in the lungs [131]. The coherence between monocyte/macrophage populations of the spleen and the liver is more comprehensible given the anatomical connection between the two organs via portal circulation [132,133]. At the same time, the spleen has been implicated as a monocyte supply for other organs as well. For instance, monocytes deposited in the spleen have been shown to migrate to inflammation foci in myocardial infarction and cerebral ischemic stroke [38,134,135].

Thus, the accumulated body of evidence suggests a general scheme of acquisition of unique properties by a tissue macrophage population. The first step involves implementation of the core macrophage differentiation program represented by PU.1, MYB, C-MAF, MAFB and ZEB2 transcription factors [136]. This universal basis becomes subsequently adjusted and refined by the influence of particular tissue niche, which shapes a transcription program characteristic of particular type of resident macrophages [39]. For KCs, the tissue niche comprises Ito cells, sinusoidal endothelial cells and hepatocytes, as well as various ECM components of hepatic parenchyma and paracrine factors; the joint influence of these components shapes a specific transcription program involving the LXR-α/ID3/SPIC expression signature in macrophages [80,111]. Interestingly, the transcriptional program of Kupffer cells turned out to be similar in mammals (humans, mice, pigs, hamsters and macaques), chicken and zebrafish [40], whereas induction of most genes unique to macrophages in these animal species required the interaction of activin receptor-like kinase (ALK1) on Kupffer cells with BMP9/10 secreted by Ito cells, and was also more or less TGFb dependent [40,111].

The concept of a macrophage niche is consistent with data on the epigenetic regulation of the resident macrophages phenotype, including those of the liver [137,138]. It has already been mentioned that the expression of the Clec4f gene, which encodes a lectin required for the presentation of alpha-galactosylceramide to natural killer T, is specific for the liver [99]. Assessment of histone modification status showed the presence of unique poised and active enhancers in the region of the Clec4f gene, as well as open chromatin regions in the region of the transcription factor LXRa, specific for Kupffer cells [138]. Histone acetylation sites were also found in the region of LXR gene, as well as in RBPJ gene, which is consistent with studies that established the dependence of the formation of a specific transcriptional program of Kupffer cells on the NOTCH-ligand DLL4 secreted by liver sinusoid endotheliocytes [111]. The effect of LPS on histone acetylation in the area of DNA regulatory regions during the differentiation of monocytes into macrophages has been also shown, which, given the constant contact with LPS, is especially important for Kupffer cells [139]. However, more detailed studies in this respect of Kupffer cells have not been conducted.

### 7.2. Interactions of Liver Stellate Cells with KCs

Liver stellate cells (Ito cells) are a component of the macrophage niche of the liver. The tissue identity of Ito cells is disputable; they are alternatively called both lipocytes and pericytes of the liver [140,141]. A characteristic feature of Ito cells is their capacity to accumulate vitamin A. Notably, similar vitamin A-accumulating cells have been identified in the lungs and the pancreas [142]. The embryonic source of Ito cells remains disputable too, although their origin can be traced to a prenatal subpopulation of WT1+ septum transversum mesenchymal cells [143]. After birth, under certain conditions (e.g., at the onset of fibrosis), new Ito cells can differentiate from mesothelial cells via epithelial-mesenchymal transition [144,145].

Ito cells implement multiple functions under both normal and pathological circumstances. Importantly, these cells synthesize ECM components, biologically active substances and signaling molecules [146], notably M-CSF which is of primary relevance to liver macrophages [80].

M-CSF exists in several forms including a secretory one found in blood plasma [147,148]. Given the variable access to the vascular bed in different organs, membrane-bound forms of M-CSF, as well as those accumulated by ECM, have special significance. The regulation of local macrophage population densities is thought to depend on M-CSF levels acting in a double-threshold mode [105]: base levels ensure macrophage survival, whereas elevated levels promote macrophage proliferation. In the event of massive death of macrophages, a local increase in M-CSF concentration takes place, caused by reduced consumption and elevated production of M-CSF by cells of the niche. This increase, overshooting the upper threshold level, ensures macrophage proliferation and eventual filling in of the vacant macrophage niches [80]. Given that M-CSF is bound to ECM or ECM-producing cells, this association will determine not just the average macrophage density but also their spatial distribution. The network of cells producing M-CSF and other factors, in conjunction with ECM ensuring local accumulation of M-CSF, have been collectively termed “nurturing scaffold” [105]. 

The topography of Kupffer cells in the perisinusoidal spaces of Disse within the liver parenchyma is almost fully identical to that of Ito cells. Along with M-CSF and IL-34 produced by themselves and ECM impregnated with these factors, Ito cells constitute the “nurturing scaffold” for the resident macrophages of the liver [80,105,149]. It should be noted that hepatic ECM is rich in growth factors other than M-CSF, e.g., HGF, participation of which in the liver macrophage functionalities is uncertain. One study shows that HGF suppresses the production of IL-6 and stimulates the synthesis of IL-10 [150]. Another already mentioned factor required for the macrophage identity determination and produced by Ito cells is BMP9/10, which binds the ALK1 receptor on KCs.

Ito cells are most likely engaged in reciprocal interactions with KCs, as macrophages have been shown to produce PDGF required for the stromal cell survival and proliferation [151]. Under pathological conditions, KCs activate the Ito cells to promote their differentiation into myofibroblasts with excessive ECM production and deposition, which eventually leads to the development of liver fibrosis [152,153]. TGFβ1, a pro-fibrogenic cytokine produced mostly by KCs, acts on Ito cells which have its cognate receptors on their surface. Under certain pathological circumstances, e.g., immunodeficiency with reduced CD4+ T cell counts within the intestinal mucosa-associated lymphoid patches, the barrier function of the intestine is compromised. As a consequence, unusually high amounts of bacterial pathogens and their derivatives start entering the liver with portal circulation, provoking a TLR4-mediated immune response [154]. Affected by the pro-inflammatory milieu, Ito cells cease to express BAMBI which normally inhibits TGFβ1-signaling, thus increasing the sensitivity of Ito cells to the TGFβ1-mediated activation and ultimately contributing to the development of liver fibrosis [155].

Upon depletion of the macrophage niche in the event of excessive macrophage cell death, Ito cells become supportive of monocyte infiltration. This transition is triggered by TNF and IL-1 massively produced by dying KCs. The activated Ito cells start expressing a plethora of monocyte chemoattractants (Ccl2, Ccl7, Cxcl10 and Pf4), as well as cell adhesion molecules (Vcam1, Sele and Icam1) responsible for the fixation of monocytes within particular niche and diapedesis [156]. Monocytes, which normally pass through liver sinusoids without delay, now stop and spread across the endothelial surface; they form processes that penetrate the sinusoidal wall and reach the space of Disse where the stellate Ito cells dwell [80]. 

Thus, the activated Ito cells are capable of transient production of chemokines that stimulate monocyte migration and promote them to express receptors that mediate cell adhesion and penetration through the endothelium. Importantly, Ito cells secrete BMP9/10 which directs the monocyte differentiation towards KC phenotypes.

### 7.3. Interactions of Sinusoidal Endothelial Cells with KCs

The liver harbors three topographically distinct populations of endothelial cells that belong, respectively, to sinusoidal capillaries, portal vessels and branches of hepatic vein the central veins drain into. Of those, only sinusoidal endothelial cells make immediate contacts with KCs, similarly to Ito cells.

Sinusoidal endothelial cells constitute about 20% of all liver cells by number, but account for less than 3% of the organ volume [157]. Sinusoidal endothelial cells come from several sources. The normal, physiological renewal occurs almost exclusively by proliferation in situ, whereas post-damage repair may involve the recruitment of bone marrow-derived precursors [158]. Sinusoidal endothelium, which continuously contacts the portal blood passing through hepatic lobules, reveals positionally defined patterns, both molecular and morphological [159]. 

Sinusoidal endothelium forms a barrier between circulation, with blood cells and plasma, and hepatic plates, with hepatocytes and stellate cells [160]. In response to VEGF (which binds VEGFR1 receptor expressed by endothelial cells) sinusoidal endothelium starts producing HGF which, in turn, stimulates hepatocyte proliferation [161]. Proliferating hepatocytes also produce angiogenic factors (VEGF, Angiopoietin 1) which promote neovascularization, thereby closing the feedback loop [162]. Thus, sinusoidal endothelium provides a key regulatory link in liver regeneration by ensuring a balance between vascular density and hepatocyte numbers [163]. In addition, sinusoidal endothelial cells have been implicated in hepatic inflammatory conditions, as they regulate the passage of macrophages and other leukocytes to hepatic plates [160,164].

In fetal development, endothelium has been shown to regulate the exit of the hepatic myeloid progenitor cell-derived monocytic precursors from the fetal liver [10] for subsequent colonization of peripheral tissues where these cells differentiate into resident macrophages [8]. PLVAP protein, which can build diaphragms in the pores of fenestrated sinusoidal endothelium in the fetal liver and interact with cell adhesion molecules, thus controlling the monocyte exit, plays a key regulatory role in this process [165]. In PLVAP-deficient adult mice, the counts of resident macrophages originating from hemopoietic lineages of the fetal liver are reduced by 95% in the peritoneum and by 70% in the spleen compared with the wild-type controls [165].

In conjunction with Ito cells, sinusoidal endothelium ensures monocyte attraction through release of chemokines and cell adhesion molecules, e.g., Vcam1, Sele, Icam1 and Selp [80]. The mechanisms of endothelial and Ito cell activation are similar and involve stimulation with TNF and IL-1 released by the dying KCs. These factors stimulate the endothelial cells to produce Bmp2 and the Notch-signaling pathways ligands Dll1 and Dll4 [80]. Experimental 21 h exposure of monocyte-derived macrophages to Dll4 in vitro promoted the expression of *Nr1h3* (LXRα) and *Spic* transcription factor genes characteristic of KCs, although it failed to activate the KC-core genes Clec4f, Cd207, Cd5l and Cdh5. The activation of the KC-core genes in the same system was achieved upon longer incubation time with BMP9 added to the culture medium. Thus, the endothelial cell DLL-Notch signaling is a strong inductor of KC identity in macrophages and its effect is further augmented by Ito cell-exerted BMP signaling [80].

### 7.4. Interactions of Hepatocytes with KCs

The liver is the biggest gland of the body and hepatocytes account for 75–80% of its volume [166]. These cells are involved in protein production and storage, carbohydrate transformations, cholesterol and bile acid salt biosynthesis and detoxification; they also participate in the innate immunity reactions [167]. Importantly, hepatocytes are constituent members of the resident macrophage niche, known to influence the transcriptomic program of KC [80]. Despite the negligible effects of liver macrophage depletion on hepatocyte transcriptomes, in vitro exposure of hepatocytes to monocyte-derived macrophages induced the latter to express transcription factor ID3, which regulates survival and determines cellular identity of KCs on a par with LXR-α [168]. Although exact molecular pathways of ID3 induction in monocytes are unknown, migration of monocytes into perisinusoidal spaces is clearly a major event enabling their contacts with hepatocytes which act as inducers [80]. In addition, hepatocytes, through production of desmosterol and on a par with Ito cells and endothelial cells, facilitate the expression of transcription factor LXR-α and its target gene module including *Cd5l, Cd38, Clec4f* and *Abca1* in monocytes [111].

## 8. Conclusions

Substantial progress in studies on the mammalian mononuclear phagocytic system has been observed over the last decade. The findings include specification of embryonic origin for various macrophage populations, as well as the role of macrophages in the regulation of organ functionalities in health, disease and regeneration. Molecular mechanisms of maintenance of the organ-specific identity of tissue macrophages remain a close focus. This is especially true for Kupffer cells of the liver due to their abundance, as well as their prominent role in the regulation and maintenance of liver functionalities. Postnatally, Kupffer cells constitute a self-renewing population replenished solely by means of local proliferation. All macrophage-monocytic lineages of the body are initially subject to the core macrophage differentiation program involving PU.1, MYB, C-MAF, MAFB and ZEB2 transcription factors. Subsequent steps of macrophage differentiation are tissue-specific: Kupffer cells, under the joint inductive influence of Ito cells, sinusoidal endothelial cells and hepatocytes, enter specific differentiation program marked by expression of LXR-α, ID3 and SPIC. The obvious progress in our understanding of macrophage functionalities opens a number of issues for further investigation. One of them is the possibility of full differentiation of monocytes, which invade the liver from circulation, into macrophages phenotypically and functionally identical to Kupffer cells; the existing evidence in this regard is controversial. Another important issue is the regulation of macrophage cell death and its role in the activation of repair processes. The role of macrophages, Kupffer cells in particular, in the immune control of the bodily functions (on a par with the nervous and endocrine systemic regulation) is still uncertain and its clarification will require dedicated efforts.

## Figures and Tables

**Figure 1 ijms-23-09868-f001:**
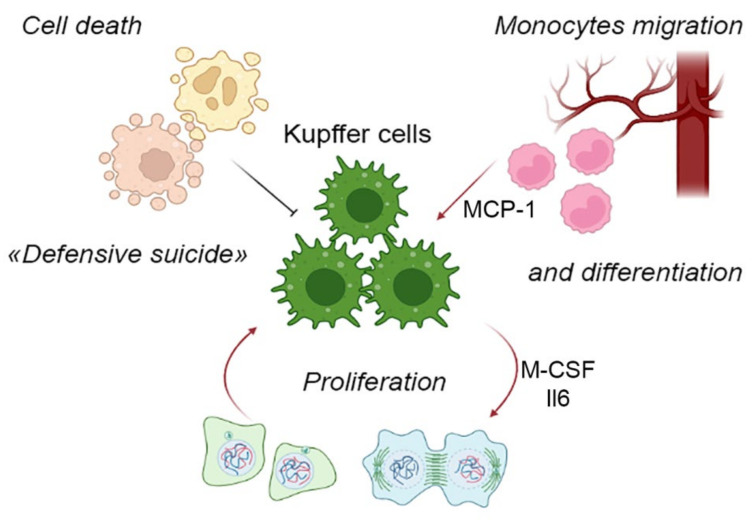
Component processes of the liver macrophage population dynamics.

**Figure 2 ijms-23-09868-f002:**
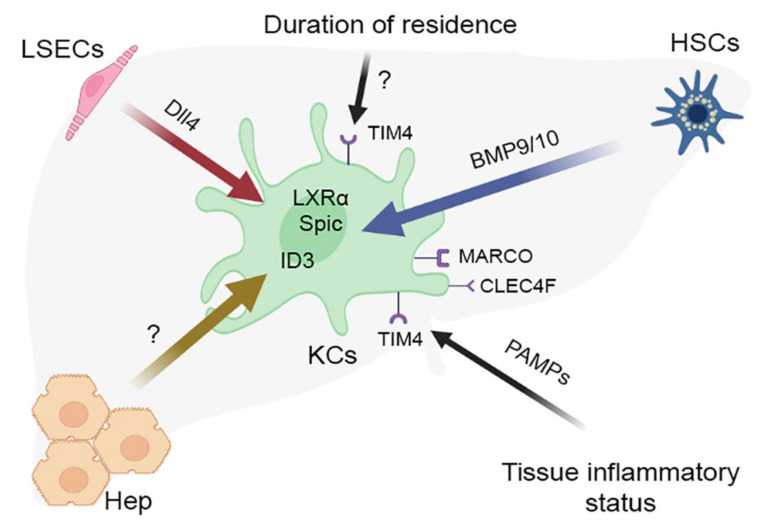
Characterization of the hepatic macrophage niche. KCs—Kupffer cells, HSCs—hepatic stellate cells (Ito cells), LSECs—liver sinusoidal endothelial cells, Hep—hepatocytes.

**Table 1 ijms-23-09868-t001:** Macrophage populations of the liver.

Population	Markers	Functions
Kupffer cells	F4/80 [58] CD68 [58] CD11b [59,60] CD163 [51,52] Cd206 (lo/hi) [53] Clec4F [61] Tim4 [61]	Homeostatic
Non-KCs Macrophages/monocytes		
Monocytes	Ly6C+ [44,50]	Inflammation
Capsule macrophages	F4/80, CD14, CD64, CD207 [62,63]	Protection against pathogens invasion from the abdominal cavity
Peritoneal macrophages	CD102, GATA6 [64]	Unclear
Biliary tree-associated macrophages	Gpnmb [41]	Unclear, lipid metabolism

## Data Availability

Not applicable.

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
