# Peer review of "An Eye on Kupffer Cells: Development, Phenotype and the Macrophage Niche"

_ijms, 2022, doi:10.3390/ijms23179868_

Round 1
Reviewer 1 Report
The review presented by the authors is very interesting but the authors should specify, focus and summarize all the aspects to be discussed, eliminating very specific aspects that can be consulted in the cited bibliography. The use of tables helps to define the objectives of each section. You should also reread the document to correct the numbering in the subsections and paragraphs. It is preferable that the MS is written in an impersonal style. In many sections the bibliographic references are absent, it is advisable to review the entire text.
Author Response
Dear Reviewer,
Thank you for appreciating our article and positive feedback. According to your recommendation in the revised version of the manuscript, we added missing references, made a new table on the populations of macrophages in the liver, corrected numbering, added the several text fragments and establish impersonal style throughout the text. We decided to consider some issues in more detail, because this allows to present the problem more clearly and it will be interesting for specialists. Hope you find the changes satisfied. Thank you!
Reviewer 2 Report
The review entitled "An eye on Kupffer cells: development, phenotype and the macrophage niche" is very well-written and highly instructive to understand biology of Kupffer cells. Authors performed interesting phenotypic comparison of KCs with others macrophages found in liver. However it should be interesting to explain how epigenetic regulation are involved in tolerogeneic response of KCs. Figures are well-designed and efficient.
Author Response
Dear Reviewer,
Thank you for appreciating and careful analysis of our Manuscript. We added the text fragment concerning epigenetic regulation are its involvement in tolerogeneic response of KCs in the revised version. Hope you find the changes satisfied!